# IMPROVED DENOISING DIFFUSION PROBABILISTIC MODELS

## ABSTRACT

We explore denoising diffusion probabilistic models, a class of generative models which have recently been shown to produce excellent samples in the image and audio domains. While these models produce excellent samples, it has yet to be shown that they can achieve competitive log-likelihoods. We show that, with several small modifications, diffusion models can achieve competitive log-likelihoods in the image domain while maintaining high sample quality. Additionally, our models allow for sampling with an order of magnitude fewer diffusion steps with only a modest difference in sample quality. Finally, we explore how sample quality and log-likelihood scale with the number of diffusion steps and the amount of model capacity. We conclude that denoising diffusion probabilistic models are a promising class of generative models with excellent scaling properties and sample quality.

## 1 INTRODUCTION

Sohl-Dickstein et al. (2015) introduced diffusion probabilistic models ("diffusion models" for brevity), a class of generative models which match a data distribution by learning to reverse a gradual, multi-step noising process. More recently, Ho et al. (2020) showed an equivalence between these models and score based generative models (Song & Ermon, 2019; 2020), which learn a gradient of the log-density of the data distribution using denoising score matching (Hyvärinen, 2005). It has recently been shown that this class of models can produce high-quality images (Ho et al., 2020; Song & Ermon, 2020; Jolicoeur-Martineau et al., 2020) and audio (Chen et al., 2020b; Kong et al., 2020), but it has yet to be shown that diffusion models can achieve competitive log-likelihoods. Furthermore, while Ho et al. (2020) showed extremely good results on the CIFAR-10 (Krizhevsky, 2009) and LSUN (Yu et al., 2015) datasets, it is unclear how well diffusion models scale to datasets with higher diversity such as ImageNet. Finally, while Chen et al. (2020b) found that diffusion models can efficiently generate audio using a small number of sampling steps, it has yet to be shown that the same is true for images.

In this paper, we show that diffusion models can achieve competitive log-likelihoods while maintaining good sample quality, even on high-diversity datasets like ImageNet. Additionally, we show that our improved models can produce competitive samples an order of magnitude faster than those from Ho et al. (2020). We achieve these results by combining a simple reparameterization of the reverse process variance, a hybrid learning objective that combines the variational lower-bound with the simplified objective from Ho et al. (2020), and a novel noise schedule which allows the model to better leverage the entire diffusion process.

We find surprisingly that, with our hybrid objective, our models obtain better log-likelihoods than those obtained by optimizing the log-likelihood directly, and discover that the latter objective has much more gradient noise during training. We show that a simple importance sampling technique reduces this noise and allows us to achieve better log-likelihoods than with the hybrid objective. Using our trained models, we study how sample quality and log-likelihood change as we adjust the number of diffusion steps used at sampling time. We demonstrate that our improved models allow us to use an order of magnitude fewer steps at test time with only a modest change in sample quality and log-likelihood, thus speeding up sampling for use in practical applications.

Finally, we evaluate the performance of these models as we increase model size, and observe trends that suggest predictable improvements in performance as we increase training compute.

## 2 DENOISING DIFFUSION PROBABILISTIC MODELS

We briefly review the formulation of diffusion models from Ho et al. (2020). This formulation makes various simplifying assumptions, such as a fixed noising process $q$ which adds diagonal Gaussian noise at each timestep. For a more general derivation, see Sohl-Dickstein et al. (2015).

### 2.1 DEFINITIONS

Given a data distribution $x_0 \sim q(x_0)$, we define a forward noising process $q$ which produces latents $x_1$ through $x_T$ by adding Gaussian noise at time $t$ with variance $\beta_t \in (0,1)$ as follows:

$$q(x_1, ..., x_T | x_0) := \prod_{t=1}^{T} q(x_t | x_{t-1}) \tag{1}$$

$$q(x_t | x_{t-1}) := \mathcal{N}(x_t; \sqrt{1 - \beta_t} x_{t-1}, \beta_t \mathbf{I}) \tag{2}$$

Given sufficiently large $T$ and a well behaved schedule of $\beta_t$, the latent $x_T$ is nearly an isotropic Gaussian distribution. Thus, if we know the exact reverse distribution $q(x_{t-1}|x_t)$, we can sample $x_T \sim \mathcal{N}(0, \mathbf{I})$ and run the process in reverse to get a sample from $q(x_0)$. However, since $q(x_{t-1}|x_t)$ depends on the entire data distribution, we approximate it using a neural network:

$$p_\theta(x_{t-1}|x_t) := \mathcal{N}(x_{t-1}; \mu_\theta(x_t, t), \Sigma_\theta(x_t, t)) \tag{3}$$

The combination of $q$ and $p$ is a variational auto-encoder (Kingma & Welling, 2013), and we can write the variational lower bound (VLB) as follows:

$$L_{\text{vlb}} := L_0 + L_1 + ... + L_{T-1} + L_T \tag{4}$$

$$L_0 := -\log p_\theta(x_0|x_1) \tag{5}$$

$$L_{t-1} := D_{KL}(q(x_{t-1}|x_t, x_0) \,||\, p_\theta(x_{t-1}|x_t)) \tag{6}$$

$$L_T := D_{KL}(q(x_T|x_0) \,||\, p(x_T)) \tag{7}$$

Aside from $L_0$, each term of Equation 4 is a $KL$ divergence between two Gaussian distributions, and can thus be evaluated in closed form. To evaluate $L_0$ for images, we assume that each color component is divided into 256 bins, and we compute the probability of $p_\theta(x_0|x_1)$ landing in the correct bin (which is tractable using the CDF of the Gaussian distribution). Also note that while $L_T$ does not depend on $\theta$, it will be close to zero if the forward noising process adequately destroys the data distribution so that $q(x_T|x_0) \approx \mathcal{N}(0, \mathbf{I})$.

It is useful to define and derive several other quantities which are relevant to the forward noising process, so we repeat them here from Ho et al. (2020):

$$\alpha_t := 1 - \beta_t \tag{8}$$

$$\bar{\alpha}_t := \prod_{s=0}^{t} \alpha_s \tag{9}$$

$$\tilde{\beta}_t := \frac{1 - \bar{\alpha}_{t-1}}{1 - \bar{\alpha}_t} \beta_t \tag{10}$$

$$\tilde{\mu}_t(x_t, x_0) := \frac{\sqrt{\bar{\alpha}_{t-1}} \beta_t}{1 - \bar{\alpha}_t} x_0 + \frac{\sqrt{\alpha_t}(1 - \bar{\alpha}_{t-1})}{1 - \bar{\alpha}_t} x_t \tag{11}$$

$$q(x_t|x_0) = \mathcal{N}(x_t; \sqrt{\bar{\alpha}_t} x_0, (1 - \bar{\alpha}_t)\mathbf{I}) \tag{12}$$

$$q(x_{t-1}|x_t, x_0) = \mathcal{N}(x_{t-1}; \tilde{\mu}(x_t, x_0), \tilde{\beta}_t \mathbf{I}) \tag{13}$$

### 2.2 TRAINING IN PRACTICE

Equation 12 provides an efficient way to jump directly to an arbitrary step of the forward noising process. This makes it possible to randomly sample $t$ during training. Ho et al. (2020) uniformly sample $t$ for each image in each mini-batch.

There are many different ways to parameterize $\mu_\theta(x_t, t)$. The most obvious option is to predict $\mu_\theta(x_t, t)$ directly with a neural network; alternatively, the network could predict $x_0$, and this output could then be fed through $\tilde{\mu}(x_t, x_0)$; finally, the network could predict the noise $\epsilon$ added to $x_0$, and this noise could be used to predict $x_0$ via

$$x_0 = \frac{1}{\sqrt{\alpha_t}} \left( x_t - \frac{\beta_t}{\sqrt{1 - \bar{\alpha}_t}} \epsilon \right) \tag{14}$$

Ho et al. (2020) found that predicting $\epsilon$ worked best, especially when combined with a reweighted loss function:

$$L_{\text{simple}} = E_{t, x_0, \epsilon} \left[ ||\epsilon - \epsilon_\theta(x_t, t)||^2 \right] \tag{15}$$

This objective can be seen as a reweighted form of $L_{\text{vlb}}$ (without the terms affecting $\Sigma_\theta$). The authors found that optimizing this reweighted objective resulted in much better sample quality than optimizing $L_{\text{vlb}}$ directly, and explain this by drawing a connection to generative score matching (Song & Ermon, 2019; 2020).

One subtlety is that $L_{\text{simple}}$ provides no learning signal for $\Sigma_\theta(x_t, t)$. This is irrelevant, however, since Ho et al. (2020) achieved their best results by fixing the variance to $\sigma_t^2 \mathbf{I}$ rather than learning it. They found that they achieve similar sample quality using either $\sigma_t^2 = \beta_t$ or $\sigma_t^2 = \tilde{\beta}_t$, which are two extremes given by $q(x_0)$ being either isotropic Gaussian noise or a delta function, respectively.

## 3 Improving the Log-likelihood

While Ho et al. (2020) found that diffusion models can generate high-fidelity samples according to FID (Heusel et al., 2017) and Inception Score (Salimans et al., 2016), they were unable to achieve competitive log-likelihoods with these models. Log-likelihood is a widely used metric in generative modeling, and it is generally believed that optimizing log-likelihood forces generative models to capture all of the modes of the data distribution (Razavi et al., 2019). Additionally, recent work (Henighan et al., 2020) has shown that small improvements in log-likelihood can have a dramatic impact on sample quality and learnt feature representations. Thus, it is important to explore why diffusion models seem to perform poorly on this metric, since this may suggest a fundamental shortcoming such as bad mode coverage. This section explores several modifications to the algorithm described in Section 2 that, when combined, allow diffusion models to achieve much better log-likelihoods on image datasets, suggesting that these models enjoy the same benefits as other likelihood-based generative models.

To study the effects of different modifications, we train fixed model architectures with fixed hyperparameters (Appendix A) on the ImageNet $64 \times 64$ (van den Oord et al., 2016a) and CIFAR-10 (Krizhevsky, 2009) datasets. While CIFAR-10 has seen more usage for this class of models, we chose to study ImageNet $64 \times 64$ as well because it provides a good trade-off between diversity and resolution, allowing us to train models quickly without worrying about overfitting. Additionally, ImageNet $64 \times 64$ has been studied extensively in the context of generative modeling (van den Oord et al., 2016b; Menick & Kalchbrenner, 2018; Child et al., 2019; Roy et al., 2020), allowing us to compare diffusion models directly to many other generative models.

The setup from Ho et al. (2020) (optimizing $L_{\text{simple}}$ while setting $\sigma_t^2 = \beta_t$ and $T = 1000$) achieves a log-likelihood of **3.99 bits/dim** on ImageNet $64 \times 64$ after 200K training iterations. We found in early experiments that we could get a boost in log-likelihood by increasing $T$ from 1000 to 4000; with this change, the log-likelihood improves to **3.77 bits/dim**. For the remainder of this section, we use $T = 4000$, but we explore this choice in Section 4.

### 3.1 Learning $\Sigma_\theta(x_t, t)$

In Ho et al. (2020), the authors set $\Sigma_\theta(x_t, t) = \sigma_t^2 \mathbf{I}$, where $\sigma_t$ is not learned. Oddly, they found that fixing $\sigma_t^2$ to $\beta_t$ yielded roughly the same sample quality as fixing it to $\tilde{\beta}_t$. Considering that $\beta_t$ and $\tilde{\beta}_t$ represent two opposite extremes, it is reasonable to ask why this choice doesn't affect samples. One clue is given by Figure 1a, which shows that $\beta_t$ and $\tilde{\beta}_t$ are almost equal except near $t = 0$, i.e. where the model is dealing with imperceptible details. Furthermore, as we increase the number of diffusion steps, $\beta_t$ and $\tilde{\beta}_t$ seem to remain close to one another for more of the diffusion process.

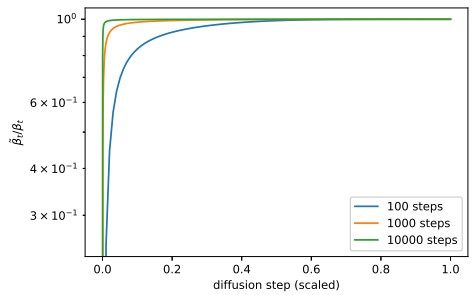 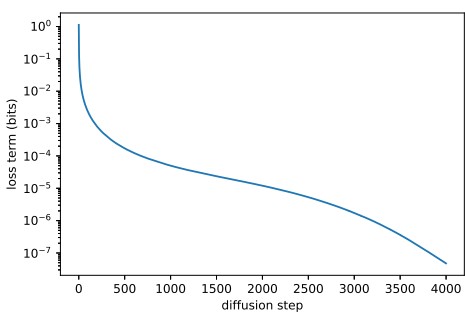

Figure 1a: The ratio $\tilde{\beta}_t/\beta_t$ for every diffusion step for diffusion processes of different lengths.

Figure 1b: Terms of the VLB vs diffusion step. The first few terms contribute most to NLL.

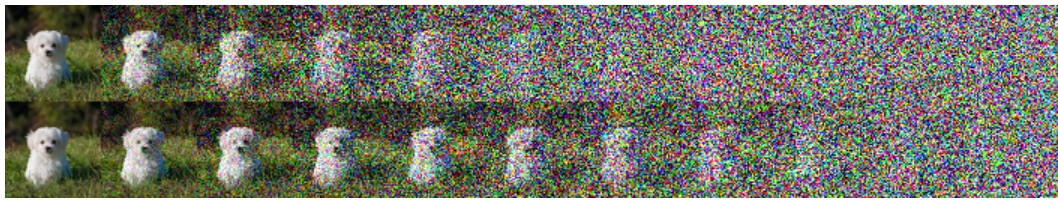

Figure 2: Latent samples from linear (top) and cosine (bottom) schedules respectively at linearly spaced values of $t$ from 0 to $T$. The latents in the last quarter of the linear schedule are almost purely noise, whereas the cosine schedule adds noise more slowly

This suggests that, in the limit of infinite diffusion steps, the choice of $\sigma_t$ might not matter at all for sample quality. In other words, as we add more diffusion steps, the model mean $\mu_\theta(x_t, t)$ determines the distribution much more than $\Sigma_\theta(x_t, t)$.

While the above argument suggests that fixing $\sigma_t$ is a reasonable choice for the sake of sample quality, it says nothing about log-likelihood. In fact, Figure 1b shows that the first few steps of the diffusion process contribute the most to the variational lower bound. Thus, it seems likely that we could improve log-likelihood by using a better choice of $\Sigma_\theta(x_t, t)$. To achieve this, we must learn $\Sigma_\theta(x_t, t)$ without the instabilities encountered by Ho et al. (2020).

Since Figure 1a shows that the reasonable range for $\Sigma_\theta(x_t, t)$ is very small, it would be hard for a neural network to predict $\Sigma_\theta(x_t, t)$ directly, even in the log domain, as observed by Ho et al. (2020). Instead, we found it better to parameterize the variance as an interpolation between $\beta_t$ and $\tilde{\beta}_t$ in the log domain. In particular, our model outputs a vector $v$ containing one component per dimension, and we turn this output into variances as follows:

$$\Sigma_\theta(x_t, t) = \exp(v \log \beta_t + (1 - v) \log \tilde{\beta}_t) \tag{16}$$

We did not apply any constraints on $v$, theoretically allowing the model to predict variances outside of the interpolated range. However, we did not observe the network doing this in practice, suggesting that the bounds for $\Sigma_\theta(x_t, t)$ are indeed expressive enough.

Since $L_{\text{simple}}$ doesn't depend on $\Sigma_\theta(x_t, t)$, we define a new hybrid objective:

$$L_{\text{hybrid}} = L_{\text{simple}} + \lambda L_{\text{vlb}} \tag{17}$$

For our experiments, we set $\lambda = 0.001$ to prevent $L_{\text{vlb}}$ from overwhelming $L_{\text{simple}}$. Along this same line of reasoning, we also apply a stop-gradient to the $\mu_\theta(x_t, t)$ output for the $L_{\text{vlb}}$ term. This way, $L_{\text{vlb}}$ can guide $\Sigma_\theta(x_t, t)$ while $L_{\text{simple}}$ is still the main source of influence over $\mu_\theta(x_t, t)$.

### 3.2 IMPROVING THE NOISE SCHEDULE

We found that the noise schedule used in Ho et al. (2020) was sub-optimal for ImageNet $64 \times 64$. In particular, the end of the forward noising process is too noisy, and so doesn't contribute very

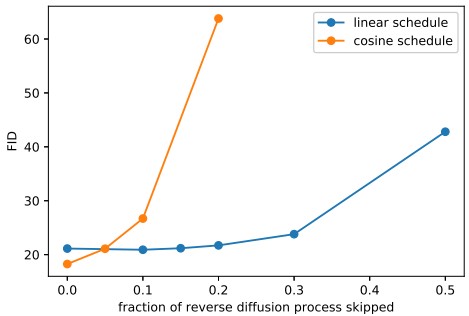

Figure 3a: FID when skipping a prefix of the reverse diffusion process on ImageNet $64 \times 64$.

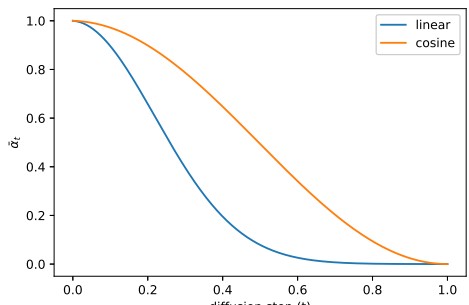

Figure 3b: $\bar{\alpha}_t$ throughout diffusion in the linear schedule and our proposed cosine schedule.

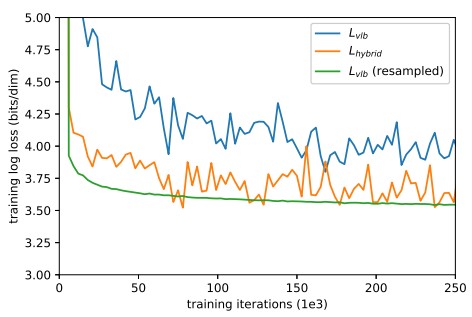

Figure 4a: Learning curves comparing the log-likelihoods achieved by different objectives on ImageNet $64 \times 64$.

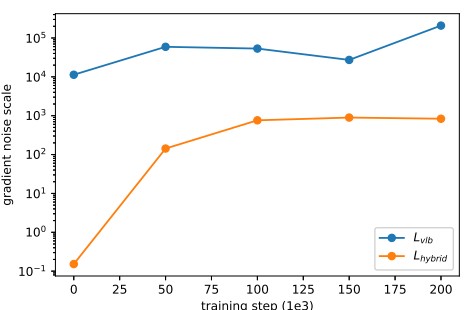

Figure 4b: Gradient noise scales for the $L_{\text{vlb}}$ and $L_{\text{hybrid}}$ objectives on ImageNet $64 \times 64$.

much to sample quality. This can be seen visually in Figure 2. The result of this effect is studied in Figure 3a, where we see that a model trained with the linear schedule does not get much worse (as measured by FID) when we skip up to 20% of the reverse diffusion process.

To address this problem, we construct a different noise schedule in terms of $\bar{\alpha}_t$:

$$\bar{\alpha}_t = \frac{f(t)}{f(0)}, \quad f(t) = \cos\left(\frac{t/T + s}{1 + s} \cdot \frac{\pi}{2}\right)^2, \quad s = 0.008 \tag{18}$$

To go from this definition to variances $\beta_t$, we note that $\beta_t = 1 - \frac{\bar{\alpha}_t}{\bar{\alpha}_{t-1}}$. In practice, we clip $\beta_t$ to be no larger than 0.999 to prevent singularities at the end of the diffusion process near $t = T$.

Our cosine schedule is designed to have a linear drop-off of $\bar{\alpha}_t$ in the middle of the process, while changing very little near the extremes of $t = 0$ and $t = T$ to prevent abrupt changes in noise level. Figure 3b shows how $\bar{\alpha}_t$ progresses for both schedules. We can see that the linear schedule from Ho et al. (2020) falls towards zero much faster, destroying information more quickly than necessary.

The small offset $s$ in our schedule prevents $\beta_t$ from being too small near $t = 0$, since we found that having tiny amounts of noise at the beginning of the process made it hard for the network to predict $\epsilon$ accurately enough. In particular, we selected $s$ such that $\sqrt{\beta_0}$ was slightly smaller than the pixel bin size, $1/127.5$. We chose to use $\cos^2$ in particular because it is a common mathematical function with the shape we were looking for. This choice was arbitrary, and we expect that many other functions with similar shapes would work as well.

### 3.3 REDUCING GRADIENT NOISE

We expected to achieve the best log-likelihoods by optimizing $L_{\text{vlb}}$ directly, rather than by optimizing $L_{\text{hybrid}}$. However, we were surprised to find that $L_{\text{vlb}}$ was actually quite difficult to optimize in practice, at least on the diverse ImageNet $64 \times 64$ dataset. Figure 4a shows the learning curves for both $L_{\text{vlb}}$ and $L_{\text{hybrid}}$. Both curves are noisy, but the hybrid objective clearly achieves better log-likelihoods on the training set given the same amount of training time.

| MODEL | TRAIN ITERS. | $T$ | SCHEDULE | OBJECTIVE | NLL (bits/dim) | FID |
|---|---|---|---|---|---|---|
| Baseline | 200K | 1K | linear | $L_{\text{simple}}$ | 3.99 | 31.0 |
| | 200K | 4K | linear | $L_{\text{simple}}$ | 3.77 | 29.7 |
| Improved | 200K | 4K | linear | $L_{\text{hybrid}}$ | 3.66 | 30.4 |
| | 200K | 4K | cosine | $L_{\text{simple}}$ | 3.68 | **25.6** |
| | 200K | 4K | cosine | $L_{\text{hybrid}}$ | 3.62 | 26.6 |
| | 200K | 4K | cosine | $L_{\text{vlb}}$ | **3.57** | 54.7 |
| Improved | 1.5M | 4K | cosine | $L_{\text{hybrid}}$ | 3.57 | **18.3** |
| | 1.5M | 4K | cosine | $L_{\text{vlb}}$ | **3.53** | 38.3 |

Table 1: Comparison of NLL and FID for different diffusion models on ImageNet $64 \times 64$. $L_{\text{vlb}}$ and $L_{\text{hybrid}}$ were trained with learned sigmas using the parameterization from Section 3.1. For $L_{\text{vlb}}$, we used the resampling scheme from Section 3.3. Using our cosine schedule and $L_{\text{hybrid}}$ improves both log-likelihood and FID over the baseline from Ho et al. (2020). Optimizing $L_{\text{vlb}}$ further improves log-likelihood at the cost of a higher FID.

| MODEL | TRAIN ITERS. | $T$ | SCHEDULE | OBJECTIVE | NLL (bits/dim) | FID |
|---|---|---|---|---|---|---|
| Baseline | 500K | 1K | linear | $L_{\text{simple}}$ | 3.73 | 3.29 |
| | 500K | 4K | linear | $L_{\text{simple}}$ | 3.37 | **2.90** |
| Improved | 500K | 4K | linear | $L_{\text{hybrid}}$ | 3.26 | 3.07 |
| | 500K | 4K | cosine | $L_{\text{simple}}$ | 3.26 | 3.05 |
| | 500K | 4K | cosine | $L_{\text{hybrid}}$ | 3.17 | 3.19 |
| | 500K | 4K | cosine | $L_{\text{vlb}}$ | **2.94** | 11.47 |

Table 2: Comparison of NLL and FID for different diffusion models on CIFAR-10. Using our cosine schedule and $L_{\text{hybrid}}$ improves log-likelihood with a marginal impact on FID. Optimizing $L_{\text{vlb}}$ further improves log-likelihood at the cost of a significantly higher FID.

We hypothesized that the gradient of $L_{\text{vlb}}$ was much noisier than that of $L_{\text{hybrid}}$. We confirmed this by evaluating the gradient noise scales (McCandlish et al., 2018) for models trained with both objectives, as shown in Figure 4b. Thus, we sought out a way to reduce the variance of $L_{\text{vlb}}$ in order to optimize directly for log-likelihood.

Noting that different terms of $L_{\text{vlb}}$ have greatly different magnitudes (Figure 1b), we hypothesized that sampling $t$ uniformly causes unnecessary noise in the $L_{\text{vlb}}$ objective. To address this, we employ importance sampling:

$$L_{\text{vlb}} = E_{t \sim p_t} \left[ \frac{L_t}{p_t} \right], \text{ where } p_t \propto \sqrt{E[L_t^2]} \text{ and } \sum p_t = 1 \qquad (19)$$

Since $E[L_t^2]$ is unknown beforehand and may change throughout training, we maintain a history of the previous 10 values for each loss term, and update this dynamically during training. At the beginning of training, we sample $t$ uniformly until we draw 10 samples for every $t \in [0, T-1]$.

With this importance sampled objective, we are able to achieve our best log-likelihoods by optimizing $L_{\text{vlb}}$.[1] This can be seen in Figure 4a as the "$L_{vlb}$ (resampled)" curve. The figure also shows that the importance sampled objective is considerably less noisy than the original, uniformly sampled objective.

## 3.4 RESULTS AND ABLATIONS

In this section, we ablate the changes we have made to achieve better log-likelihoods. Table 1 summarizes the results of our ablations on ImageNet $64 \times 64$, and Table 2 shows them for CIFAR-10. We also trained our best ImageNet $64 \times 64$ models for 1.5M iterations, and report these results as well. Based on the results, we recommend always using the cosine schedule, and the $L_{\text{hybrid}}$ objective in most cases. If one is only optimizing for likelihood and not sample quality, the importance sampled $L_{\text{vlb}}$ is the best objective to use.

---

[1]We found that the importance sampling technique was not helpful when optimizing $L_{\text{hybrid}}$ directly.

| MODEL | ImageNet $64 \times 64$ NLL (bits/dim) | CIFAR-10 NLL (bits/dim) |
|---|---|---|
| Glow (Kingma & Dhariwal, 2018) | 3.81 | 3.35 |
| Flow++ (Ho et al., 2019) | 3.69 | 3.08 |
| PixelCNN (van den Oord et al., 2016b) | 3.57 | 3.14 |
| PixelSNAIL (Chen et al., 2018) | 3.52 | 2.85 |
| SPN (Menick & Kalchbrenner, 2018) | 3.52 | - |
| Image Transformer (Parmar et al., 2018) | 3.48 | 2.90 |
| Sparse Transformer (Child et al., 2019) | 3.44 | **2.80** |
| Routing Transformer (Roy et al., 2020) | **3.43** | - |
| Diffusion (Ho et al., 2020) | 3.77 | 3.70 |
| Improved Diffusion (ours) | **3.53** | **2.94** |

Table 3: Comparison of diffusion models to other likelihood-based models on CIFAR-10 and Unconditional ImageNet $64 \times 64$. On ImageNet $64 \times 64$, our model is competitive with the best conventional models, but is worse than fully transformer-based architectures.

## 4  IMPROVING SAMPLING SPEED

All of our models were trained with 4000 diffusion steps, and thus producing a single sample takes several minutes on a modern GPU. In this section, we explore how performance scales if we reduce the steps used during sampling, and find that our pre-trained models can produce high-quality samples with many fewer diffusion steps than they were trained with without any fine-tuning. Reducing the steps in this way makes it possible to sample from our models in a number of seconds rather than minutes, and greatly improves the practical applicability of image diffusion models.

For a model trained with $T$ diffusion steps, we would typically sample using the same set of $t$ values $(1, 2, ..., T)$ as used during training. However, it is also possible to sample using an arbitrary set of $t$ values. We define a sequence $S$ of $t$ values to use for sampling, such as a strided schedule like $S = (1, 3, 5, ..., T - 1)$. Given the training noise schedule $\bar{\alpha}_t$, we can obtain the sampling noise schedule $\bar{\alpha}_{S_t}$, which can be used to obtain corresponding sampling variances

$$\beta_{S_t} = 1 - \frac{\bar{\alpha}_{S_t}}{\bar{\alpha}_{S_{t-1}}}, \quad \tilde{\beta}_{S_t} = \frac{1 - \bar{\alpha}_{S_{t-1}}}{1 - \bar{\alpha}_{S_t}} \beta_{S_t} \tag{20}$$

We can compute $p(x_{S_{t-1}}|x_{S_t})$ as $\mathcal{N}(\mu_\theta(x_{S_t}, S_t), \Sigma_\theta(x_{S_t}, S_t))$. Note that $\Sigma_\theta(x_{S_t}, S_t)$ is parameterized as a range between $\beta_{S_t}$ and $\tilde{\beta}_{S_t}$ so it will automatically be rescaled for the shorter diffusion process.

To evaluate sample quality for reduced numbers of sampling steps, we use a stride $K$ over timesteps to reduce the total number of sampling steps from $T$ to $T/K$. In Figures 5a and 5c, we evaluate FIDs for an $L_{\text{hybrid}}$ model and an $L_{\text{simple}}$ model that were trained with 4000 diffusion steps, using 30, 50, 100, 150, 200, 400, and 4000 sampling steps. We do this for multiple checkpoints throughout training. We find that the $L_{\text{simple}}$ model suffers much more in sample quality when using a reduced number of sampling steps, whereas our $L_{\text{hybrid}}$ model maintains sample quality. Furthermore, we find that using more sampling steps becomes increasingly beneficial throughout training. However, 100 sampling steps is still sufficient to achieve near-optimal FIDs for our fully trained models.

In initial experiments, we found that although constant striding did not significantly affect FID, it drastically reduced log-likelihood. To address this, we use a strided subset of timesteps as for FID (with stride $K$), but we also include every $t$ from 1 to $T/K$. This requires $T/K$ extra evaluation steps, but greatly improves log-likelihood compared to the uniformly strided schedule. In Figures 5b and 5d we present log-likelihoods with this modified strided schedule.

## 5  SCALING MODEL SIZE

In the previous sections, we showed algorithmic changes that improved log-likelihood and FID without changing the amount of training compute. However, a trend in modern machine learning is that larger models and more training time tend to improve model performance (Kaplan et al., 2020;

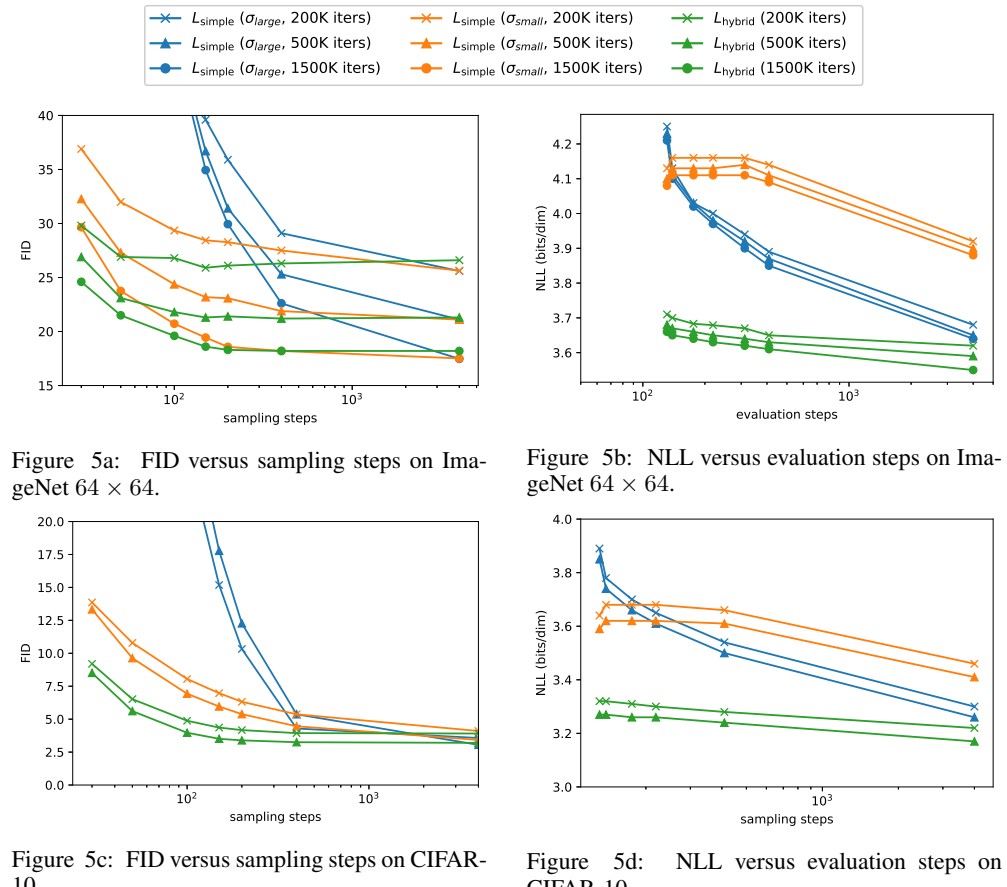

Figure 5a: FID versus sampling steps on ImageNet $64 \times 64$.

Figure 5b: NLL versus evaluation steps on ImageNet $64 \times 64$.

Figure 5c: FID versus sampling steps on CIFAR-10.

Figure 5d: NLL versus evaluation steps on CIFAR-10.

Figure 5: NLL and FID versus number of evaluation/sampling steps, for models trained on ImageNet $64 \times 64$ and CIFAR-10. All models were trained with 4000 diffusion steps. Models that learn sigmas using our reparametrization and $L_{\text{hybrid}}$ objective (Section 3.1) increase marginally in NLL and FID as we reduce evaluation/sampling steps, while using fixed sigmas as in Ho et al. (2020) results in a larger increase.

Chen et al., 2020a; Brown et al., 2020). Given this observation, we investigate how FID and NLL scale as a function of model size. Our results suggest that diffusion models can achieve better and better performance as training compute increases.

To measure how performance scales with compute, we train four different models on ImageNet $64 \times 64$ with the $L_{\text{hybrid}}$ objective described in Section 3.1. To change model capacity, we apply a depth multiplier across all layers, such that the first layer has either 64, 96, 128, or 192 channels. Note that our previous experiments used 128 channels in the first layer. Since the depth of each layer affects the scale of the initial weights, we scale the Adam learning rate for each model by $1/\sqrt{\text{channel multiplier}}$, such that the 128 channel model has a learning rate of 0.0001 (as in our other experiments).

Figure 6a and 6b show how FID and NLL improve relative to compute. These plots reveal that, to achieve optimal performance for a given amount of compute, it often makes sense to train a larger model for fewer iterations, rather than training a smaller model to convergence. We note that these models do not achieve optimal log-likelihoods because they were trained with our $L_{\text{hybrid}}$ objective and not directly with $L_{\text{vlb}}$ to keep both good log-likelihoods and sample quality. The x-axis in both figures is the theoretical amount of training compute, assuming full hardware utilization.

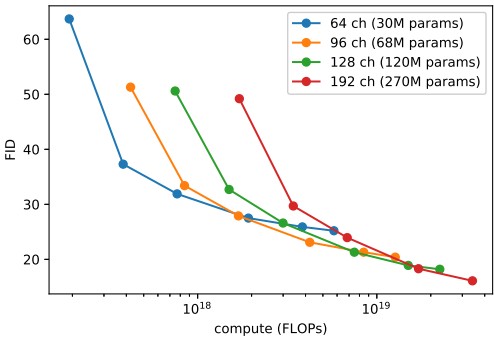 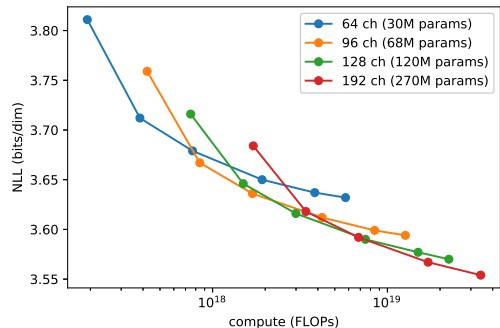

Figure 6a: FID throughout training on ImageNet $64 \times 64$ for different model sizes.

Figure 6b: NLL throughout training on ImageNet $64 \times 64$ for different model sizes.

## 6 RELATED WORK

Chen et al. (2020b) and Kong et al. (2020) are two recent works that use diffusion models to produce high fidelity audio conditioned on mel-spectrograms. Concurrent to our work, Chen et al. (2020b) use a combination of improved schedule and $L_1$ loss to allow sampling with fewer steps with very little reduction in sample quality. However, compared to our unconditional image generation task, their generative task has a strong input conditioning signal provided by the mel-spectrograms, and we hypothesize that this makes it easier to sample with fewer diffusion steps.

Jolicoeur-Martineau et al. (2020) explored score matching in the image domain, and constructed an adversarial training objective to produce better $x_0$ predictions. However, they found that choosing a better network architecture removed the need for this adversarial objective, suggesting that the adversarial objective is not necessary for powerful generative modeling.

## 7 CONCLUSION

We have shown that, with a few modifications, diffusion models can sample much faster and achieve better log-likelihoods with little impact on sample quality. Here we summarize our main findings:

- Our cosine noise schedule improves NLL (and sometimes FID) compared to the linear schedule from Ho et al. (2020).
- Learning $\Sigma_\theta$ using our parameterization and $L_{\text{hybrid}}$ objective provides a good trade-off between NLL and FID. More importantly, it allows sampling with many fewer steps without decreased sample quality.
- One can optimize $L_{\text{vlb}}$ directly using our importance sampling technique to achieve the best possible NLL at the expense of sample quality.

The combination of these results makes diffusion models an attractive choice for generative modeling, since they combine good log-likelihoods, high-quality samples, and fast sampling with a well-grounded, stationary training objective. Furthermore, we have investigated how diffusion models scale with the amount of available training compute, and found that more training compute trivially leads to better sample quality and log-likelihood. These results indicate that diffusion models are a promising direction for future research, especially as the affordability of compute increases over time.

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

# A  HYPERPARAMETERS

For all of our experiments, we use a UNet model architecture[2] similar to that used by Ho et al. (2020). We changed the attention layers to use multi-head attention (Vaswani et al., 2017), and opted to use four attention heads rather than one (while keeping the same total number of channels). We employed attention not only at the 16x16 resolution, but also at the 8x8 resolution. Additionally, we changed the way the model conditions on $t$. In particular, instead of computing conditioning vector $v$ and injecting it into hidden state $h$ as $\mathrm{GroupNorm}(h+v)$, we compute conditioning vectors $w$ and $b$ and inject them into the hidden state as $\mathrm{GroupNorm}(h)(w+1)+b$. We found in preliminary experiments on ImageNet $64 \times 64$ that these modifications slightly improved FID.

We used a 120M parameter model for all ImageNet $64 \times 64$ experiments except in Section 5, where we scaled the number of channels in all layers. In this architecture, the downsampling stack performs four steps of downsampling, each with three residual blocks (He et al., 2015). The upsampling stack is setup as a mirror image of the downsampling stack. From highest to lowest resolution, the UNet stages use $[C, 2C, 3C, 4C]$ channels, respectively. In all experiments except those in Section 5, we set $C = 128$. We estimate that, with $C = 128$, our model requires roughly 39 billion FLOPs in the forward pass.

For our CIFAR-10 experiments, we used a smaller model with three resblocks per downsampling stage and layer widths $[C, 2C, 2C, 2C]$ with $C = 128$. We swept over dropout values $\{0.1, 0.2, 0.3\}$ and found that 0.1 worked best for the linear schedule while 0.3 worked best for our cosine schedule (Section 3.2). We expand upon this in Appendix E.

For all of our experiments, we used Adam (Kingma & Ba, 2014) with a batch size of 128 and an exponential moving average (EMA) over model parameters with a rate of 0.9999. Except in Section 5, we fixed the learning rate to $0.0001$. For quick comparisons in Section 3, we trained models for 200K iterations. This is not enough to reach convergence, but we believe it is enough to fairly compare different modifications. We then trained the best models for 1.5M iterations to achieve better performance.

When using the linear noise schedule from Ho et al. (2020), we linearly interpolated from $\beta_1 = 0.0001/4$ to $\beta_{4000} = 0.02/4$ in order to preserve the shape of $\bar{\alpha}_t$ for the $T = 4000$ schedule.

When computing FID for CIFAR-10, we produce 50K samples and compare them against the training set for consistency with other work. When computing FID for ImageNet $64 \times 64$, we produce 10K samples and compute FID against 50K validation images unless otherwise stated. Using only 10K samples biases the FID to be worse-than-necessary, but requires much less compute for sampling. Since we mainly use FID for relative comparisons, this bias is acceptable.

---

[2]In initial experiments, we found that a ResNet-style architecture with no downsampling achieved better log-likelihoods but worse FIDs than the UNet architecture.

# B SAMPLES

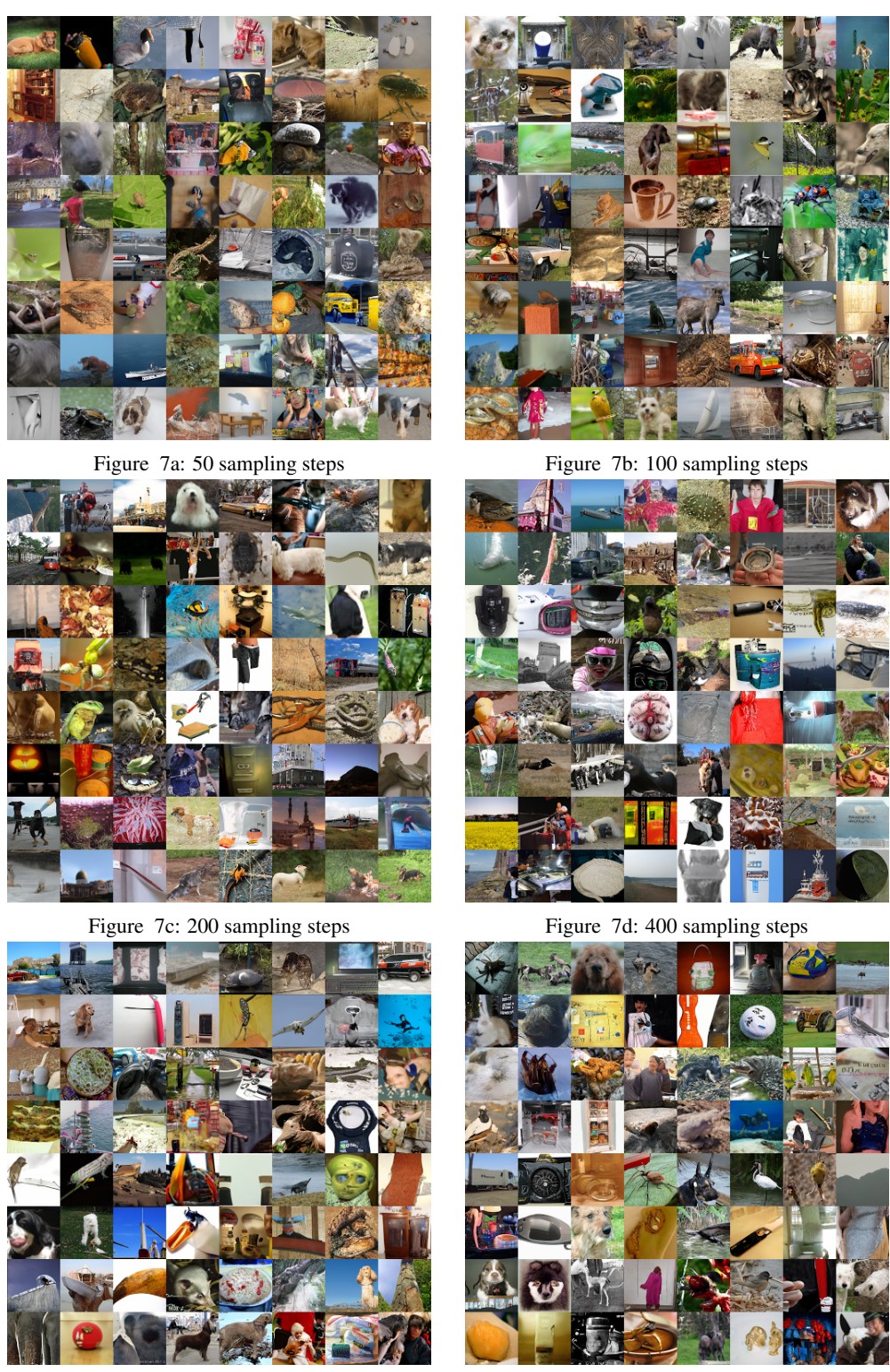

Figure 7a: 50 sampling steps

Figure 7b: 100 sampling steps

Figure 7c: 200 sampling steps

Figure 7d: 400 sampling steps

Figure 7e: 1000 sampling steps

Figure 7f: 4000 sampling steps

Figure 7: Unconditional ImageNet $64 \times 64$ samples as we reduce number of sampling steps for a $L_{\text{hybrid}}$ model with $4K$ diffusion steps trained for 1.5M training iterations.

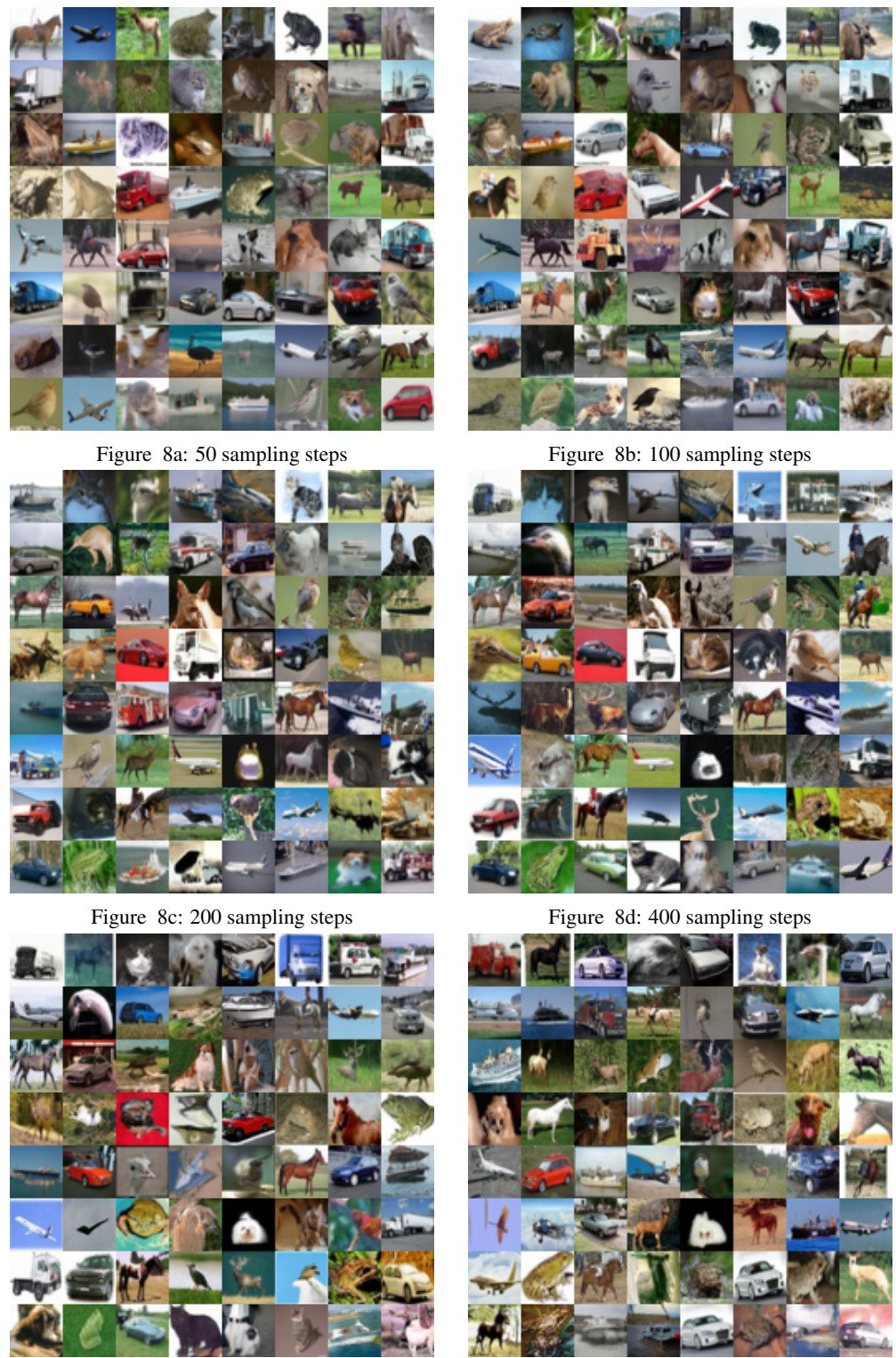

Figure 8a: 50 sampling steps

Figure 8b: 100 sampling steps

Figure 8c: 200 sampling steps

Figure 8d: 400 sampling steps

Figure 8e: 1000 sampling steps

Figure 8f: 4000 sampling steps

Figure 8: Unconditional CIFAR-10 samples as we reduce number of sampling steps for a $L_{\text{hybrid}}$ model with $4K$ diffusion steps trained for 500K training iterations.

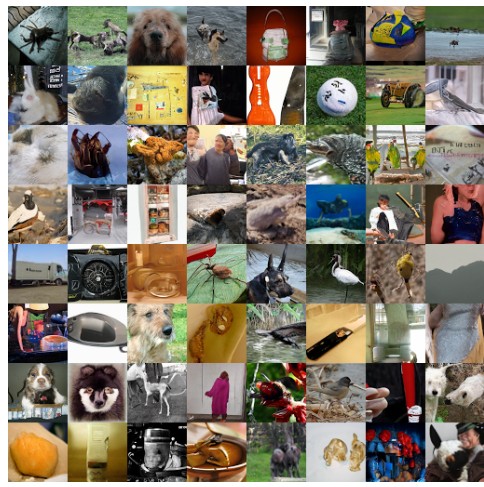 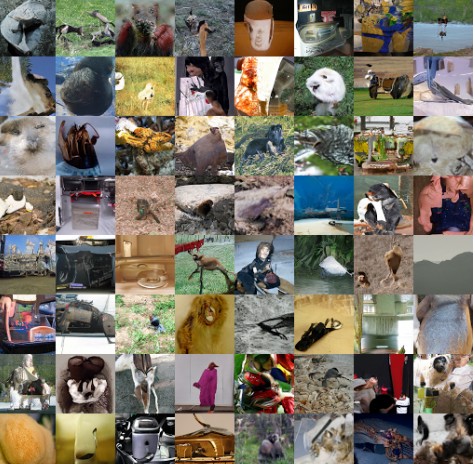

Figure 9a: Samples from $L_{\text{hybrid}}$ model          Figure 9b: Samples from $L_{\text{vlb}}$ model

Figure 9: Unconditional ImageNet $64 \times 64$ samples generated from an $L_{\text{hybrid}}$ and $L_{\text{vlb}}$ model respectively using the exact same random noise. Both models were trained for 1.5M training iterations.

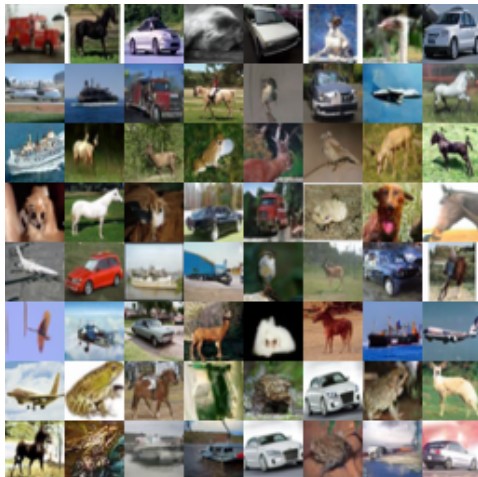 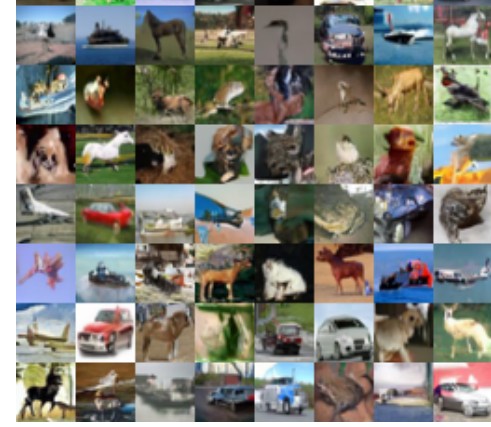

Figure 10a: Samples from $L_{\text{hybrid}}$ model          Figure 10b: Samples from $L_{\text{vlb}}$ model

Figure 10: Unconditional CIFAR-10 samples generated from an $L_{\text{hybrid}}$ and $L_{\text{vlb}}$ model respectively using the exact same random noise. Both models were trained for 500K training iterations.

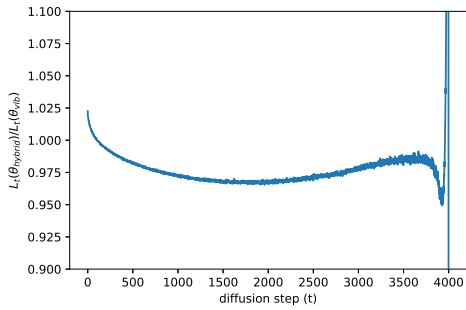

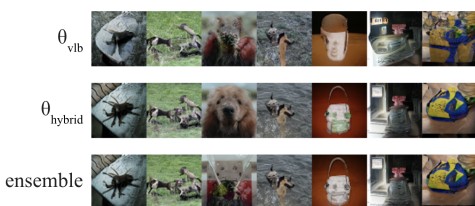

Figure 11a: The ratio between VLB terms for each diffusion step of $\theta_{\text{hybrid}}$ and $\theta_{\text{vlb}}$. Values less than 1.0 indicate that $\theta_{\text{hybrid}}$ is "better" than $\theta_{\text{vlb}}$ for that timestep of the diffusion process.

Figure 11b: Samples from $\theta_{\text{vlb}}$ and $\theta_{\text{hybrid}}$, as well as an ensemble produced by using $\theta_{\text{vlb}}$ for the first and last 100 diffusion steps. For these samples, the seed was fixed, allowing a direct comparison between models.

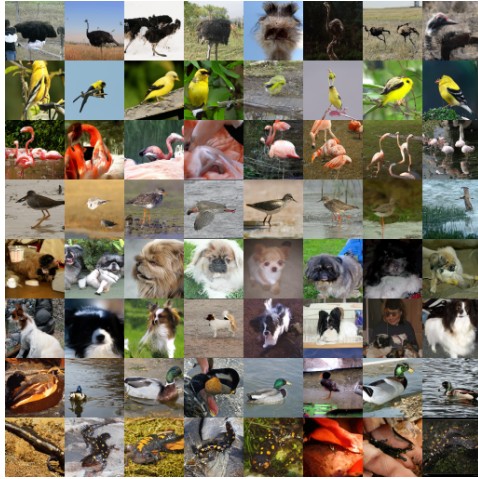

Figure 12a: Samples with random noise.

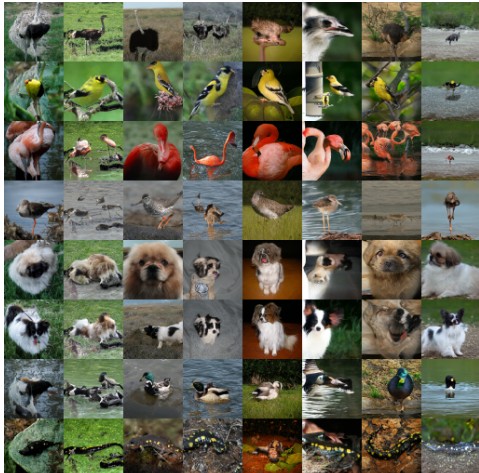

Figure 12b: Samples with same noise in a column

Figure 12: Conditional ImageNet $64 \times 64$ samples generated from an $L_{\text{hybrid}}$ model trained for 1.7M training steps. The classes are 9: ostrich, 11: goldfinch, 130: flamingo, 141: redshank, 154: pekinese, 157: papillon, 97: drake and 28: spotted salamander. On right we fix the random noise seed in each column to see how the class label affects the sampling process.

## C    COMBINING $L_{\text{HYBRID}}$ AND $L_{\text{VLB}}$ MODELS

To understand the trade-off between $L_{\text{hybrid}}$ and $L_{\text{vlb}}$, we show in Figure 11a that the model resulting from $L_{\text{vlb}}$ (referred to as $\theta_{\text{vlb}}$) is better at the start and end of the diffusion process, while the model resulting from $L_{\text{hybrid}}$ (referred to as $\theta_{\text{hybrid}}$) is better throughout the middle of the diffusion process. This suggests that $\theta_{\text{vlb}}$ is focusing more on imperceptible details, hence the lower sample quality.

Given the above observation, we performed an experiment on ImageNet $64 \times 64$ to combine the two models by constructing an ensemble that uses $\theta_{\text{hybrid}}$ for $t \in [100, T - 100)$ and $\theta_{\text{vlb}}$ elsewhere. We found that this model achieved an FID of **18.9** and an NLL of **3.52 bits/dim**. As we see from Table 1, this is only slightly worse than $\theta_{\text{hybrid}}$ in terms of FID, while being better than both models in terms of NLL.

## D    COMPARING SAMPLE QUALITY TO OTHER GENERATIVE MODELS

While this paper does not focus on comparing sample quality to other types of generative models, we were curious how diffusion models compared to modern generative models on ImageNet $64 \times 64$. Unfortunately, we did not find any literature which computed FID for unconditional ImageNet

| MODEL | FID |
|---|---|
| FQ-GAN (Zhao et al., 2020) | 9.67 |
| Instance Selection GAN (DeVries et al., 2020) | 9.07 |
| Improved Diffusion (ours) | **8.43** |

Table 4: Sample quality comparison on class conditional ImageNet $64 \times 64$.

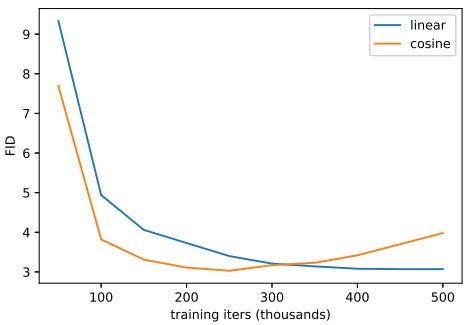

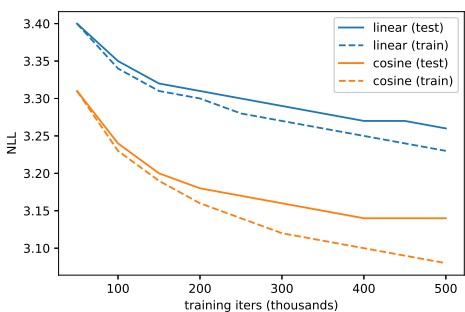

Figure 13a: FID over the course of training.

Figure 13b: Negative log-likelihood over the course of training.

Figure 13: Evaluation metrics over the course of training for two CIFAR-10 models, both with dropout 0.1. The model trained with the linear schedule learns more slowly, but does not overfit as quickly. When too much overfitting occurs, we observed overfitting artifacts similar to those from Salimans et al. (2017), which is reflected by increasing FID.

$64 \times 64$. We ran an additional experiment where we trained a class-conditional diffusion model for 1.7M iterations using the $L_{\text{hybrid}}$ objective. To make the model class-conditional, we inject class information through the same pathway as the timestep $t$. In particular, we add a class embedding $v_i$ to the timestep embedding $e_t$, and pass this embedding to residual blocks throughout the model. When computing FID for this task, we generated 50K samples (rather than 10K) to be directly comparable to other works. We found that using more samples led to a decrease in estimated FID of roughly 2 points. This is the only FID we report that was computed using 50K samples. Figure 12 shows our samples, and Table 4 summarizes our results.

## E   OVERFITTIG ON CIFAR-10

On CIFAR-10, we noticed that all models overfit, but tended to reach similar optimal FID at some point during training. Holding dropout constant, we found that models trained with our cosine schedule tended to reach optimal performance (and then overfit) more quickly than those trained with the linear schedule (Figure 13). In our experiments, we corrected for this difference by using more dropout for our cosine models than the linear models. We suspect that the overfitting from the cosine schedule is either due to 1) less noise in the cosine schedule providing less regularization, or 2) the cosine schedule making optimization, and thus overfitting, easier.

