# OpenReview forum: "Improved Denoising Diffusion Probabilistic Models"
_ICLR.cc/2021/Conference — Reject_

### Official Review · AnonReviewer3 · 2020-10-26
**Tricks to improve log likelihood of diffusion models while maintain their sample quality**

**Rating:** 5
**Confidence:** 2

**Review:**

The paper found several methods to improve log likelihood of diffusion models while maintain their sample quality, including cosine instead of linear noise schedule, using a hybrid objective to learn parameters of the covariance function, and using importance sampling to improve the gradient noise. The authors also explore how sample quality and log likelihood scale with the number of diffusion steps and model capacity. Experiments on 64x64 ImageNet dataset show competitive llh while keeping the sample quality.

Clarity: as the denoising diffusion model, especially its success in generating high-quality image samples, is still quite new, it would be beneficial if the paper could describe the technical background of it and the existing variational training methods in more details. Section 2 kind of serves this purpose, but it misses many important steps and focuses more on defining terms used in later sections.

Significance of this work: The necessity for having larger log likelihood for the diffusion model is not very well motivated. If the model is mostly used to generate high-quality samples and we have known how to train it to do so, why does the LLH values still matter?

Originality: using cosine noise schedule, new parameterization and hybrid objective seems effective for training, but doesn't seem to be very innovative. But I'm not very familiar with the denoising diffusion model and could be wrong. The results on how the model scales with computation seems trivial and may have been known already. Finally, it'll be beneficial if the authors can verify findings got in this paper can apply to other dataset or types of data more broadly.

---

> ### Author Response · Authors · 2020-11-20
> **Added CIFAR-10 results, argue for significance (fast sampling is important!), inquire about clarity.**
>
> Thanks for taking the time to read our paper and provide very insightful feedback!
>
> You make an interesting point about the background section. It’s hard to find a good balance between brevity and richness in these types of overviews. We were aiming to cover the necessary notation while keeping the reader out of the weeds, since the math behind diffusion models is rich and most of it is unnecessary to understand our key contributions. However, we will be happy to add more discussion to this section to clarify concepts which are left unclear. Are there any points of confusion in particular that you feel we should address in this section?
>
> Regarding significance: while we do focus on achieving better log-likelihood, our most impactful result is that learning sigma--which happens to improve log-likelihood--also improves sample quality when sampling with fewer diffusion steps. This >10x sampling speedup is a very important result, because it makes diffusion models much more practically applicable.
>
> While there is some amount of intrinsic benefit to achieving a better likelihood (i.e. a better likelihood means better compression algorithms), there are some secondary benefits to advancing this metric. Log-likelihood is a commonly used metric throughout generative modeling literature, and it is useful to be able to compare models along this metric. Typically, likelihood-based models tend to have better mode coverage than GANs, and the previously poor log-likelihoods of diffusion models signalled to researchers that diffusion models might fall into the same pitfalls as GANs (e.g. bad mode-coverage). By achieving better log-likelihood with a tiny change, we have hopefully convinced researchers that diffusion models are effective likelihood-based models and merit further research.
>
> Our scaling result was not previously known to our knowledge. One non-trivial takeaway from the scaling curves is that, given a compute budget, it is optimal to train a larger model for fewer iterations, rather than training a smaller model to convergence. It might seem trivial that using a wider network will achieve better sample quality in general, but this has not been the case for other popular ideas in ML which worked on small scales and didn’t scale favorably (e.g. early VAEs, EBMs, SVMs, Gaussian Processes). Many generative models, like GANs, are non-trivial to scale without additional tricks. GANs in particular present various difficulties when scaling, since training for longer doesn’t necessarily mean better samples (e.g. BigGAN found that training diverged after a certain number of steps for all models).
>
> We agree that it is important to show results on other datasets for a wide variety of reasons. To address this, we have released CIFAR-10 results in the next revision of our paper.

---

### Official Review · AnonReviewer2 · 2020-10-26
**Official Blind Review #2**

**Rating:** 5
**Confidence:** 3

**Review:**

Denoising diffusion probabilistic models have been proved to produce excellent samples in the image and audio domains. However, it has yet to be shown that they can achieve competitive log-likelihoods. This paper shows that with several small modifications, diffusion models can achieve competitive log-likelihoods in the image domain while maintaining high sample quality. This paper is well-written and good-organized. However, I have the following concerns.

1.	The authors claim that the noise schedule used in Ho et al. (2020) was, experimentally, sub-optimal for ImageNet $64 \times 64$, which lacks theoretical guarantees.
2.	This manuscript is mainly based on the previous work Ho et al. (2020). The novelty seems to be too limited.
3.	I am not convinced that only one dataset (ImageNet $64 \times 64$) is sufficient to demonstrate the performance of the proposed strategy.

---

> ### Author Response · Authors · 2020-11-20
> **Added CIFAR-10 dataset; elaborate on points of novelty.**
>
> Thanks for taking the time to read through our paper and leave feedback!
>
> 1. We agree that we should expand upon our motivation for the cosine schedule, and have gathered more experiments (i.e. CIFAR-10) to argue for its empirical success. Regarding your point, what sorts of theoretical guarantees are you referring to? Both our schedule and the one in Ho et al. are fairly ad-hoc and based on the fact that they produce good results, empirically.
>
> 2. While our paper builds heavily on Ho et al., we do make a number of significant contributions. First, we show that diffusion models can produce samples an order of magnitude more efficiently with a small change in the training procedure. Additionally, we show that diffusion models can achieve competitive log-likelihoods, which is a compelling argument for why researchers should invest time into them. In particular, achieving good log-likelihood is a sign of good mode coverage, and without such a result, many researchers were skeptical of diffusion models’ ability to achieve good mode coverage. Finally, we investigate and resolve an issue where optimizing a theoretically-sound objective, $L_{vlb}$, doesn’t yield optimal log-likelihoods as one might expect. We believe that all of these contributions are novel and useful.
>
> 3. Agreed that multiple datasets will be helpful for making our point. We have released CIFAR-10 results in the latest revision of our paper, since it is a common dataset used throughout other recent papers on this class of models. We hope that this will help demonstrate the generality of our findings.

---

### Official Review · AnonReviewer4 · 2020-10-29
**The authors explore behavior of likelihoods for diffusion models. Some useful experiments. Need work.**

**Rating:** 5
**Confidence:** 4

**Review:**

The paper talks builds upon the recent work from Ho (2020) about generative models that use noise diffusion. The authors suggest that the proposal in Ho can not only be used in good quality sample generation (as already shown by Ho), but also leads to reasonable improvements in likelihood. Overall, some of the ideas presented in the paper are interesting and useful; but the paper overall needs work.

Other questions/concerns:
1. Firstly, from an application point of view, what does achieving a high log-likelihood mean, if the samples are already good enough or high quality?
2. How do we interpret the bits/dim metric here? Its rather hard to rationalize that a change in 0.01 makes sense in this metric? And more generally, what are we aiming for in terms of a reasonable change?
3. In section 3.1; how did we end up needing to tune big-sigma_theta (x_t, t) while arguing that fixing small-sig_t^2 is ok? This is in section 3.1 second paragraph; Either I am missing something of the argument here is that we need to tune noise variance and cannot fix it?
4. What is the intuition behind expecting the (squared) cosine schedule to work? It is interesting to think that a periodic decay noising schedule is better than a linear one?
5. And related to that, do not understand this weird value of 0.008 for s? The whole point here is some small non-zero s is ok; why specifically 0.008?!
6. One of the main conclusions in section 3.4 is kind of confusing --- based on the summary, if we are not interested in sample quality but only interested in maxing of likelihood, then the proposal of this work is not good, and working with L_vlb suffices? Is this correct? Based on the motivation, it seems the opposite was being claimed i.e., L_hybrid is important for maxing of likelihood (third para in introduction)?

---

> ### Author Response · Authors · 2020-11-20
> **Justified/cited importance of log-likelihood, clarify various points in paper**
>
> First, we would like to thank you for taking the time to provide your helpful feedback! Your review has shown us various changes and clarifications we should make to our paper.
>
> 1. We chose to focus on log-likelihood because it is a commonly-used metric throughout generative modeling literature. Typically, good log-likelihood is a sign of good mode coverage, and without such a result, many researchers were skeptical of diffusion models’ ability to achieve good mode coverage. A theoretical motivation for log-likelihood is that better log-likelihoods mean better data compression, although deep learning models are not often used in modern compression algorithms (so far!) due to performance. We have added more motivation for log-likelihood to the beginning Section 3.
>
> 2. Prior work (e.g. https://arxiv.org/abs/2010.14701) has shown that, when the log-likelihood is close to the intrinsic entropy of the distribution, tiny improvements in log-likelihood can lead to drastic changes in sample quality and learned representations. Given that all recent state-of-the-art models land in the range of 3.4-3.6 bits/dim for ImageNet 64x64, it seems safe to assume that 0.01 bits/dim could actually be a meaningful improvement.
>
> 3. Sorry for the confusion! We argued that fixing sigma is okay when we use many diffusion steps and don’t care about log-likelihood. However, we then go on to argue (and show) that learning sigma is helpful in two cases: 1) when sampling with fewer diffusion steps, 2) when we want lower log-likelihood. The first point is especially important, because it means that learning sigma allows you to use much less compute for sampling.
>
> 4. Agreed that we should elaborate more in the paper on how we designed the cosine schedule. One of our takeaways from Ho et al. was that their linear schedule was hand-crafted and has two free hyperparameters, and the lack of motivation for its exact mathematical form hinted that the schedule was low-hanging fruit for improvement. We crafted our cosine schedule with a few goals in mind that we hoped would improve results. First, we wanted a schedule that added noise more slowly than the linear schedule, as can be seen in Figure 3b, but we also wanted it to be flat near t=0 and t=1 to avoid abrupt changes in noise level. We chose cosine^2 since it has these properties and uses a standard mathematical function; we then added coefficients so that the schedule was monotonic in the range [0,1]. Due to our coefficients, the schedule is actually not periodic, but monotonic.
>
> 5. We have updated our paper to explain our choice of settings more carefully, since we agree that it was unclear. To clarify briefly, we chose s such that $\sqrt{\beta_0}$ was slightly smaller than the pixel bin size 1/127.5. This allows the model to narrow in on an individual pixel (helping log-likelihood and smoothness of the generated images). We found that making the first few $\beta$ values too small (e.g. with s=0) resulted in very high loss terms near t=0, likely due to numerical issues where the model is expected to predict discrete pixel values too accurately.
>
> 6. There are multiple objectives one might wish to optimize for. If you strictly want to use this model in some type of compression algorithm, then using L_vlb is the best choice. Though directly optimizing L_vlb doesn’t work due to gradient noise, we show that we can make it work much better using importance sampling (Section 3.3). However, most of the time you care more about fast sampling and sample quality. In this case, L_hybrid is the best choice. One key takeaway from this paper is that, even though L_hybrid doesn’t achieve optimal log-likelihoods, it greatly improves sample quality when sampling with fewer steps.

---

### Official Review · AnonReviewer5 · 2020-11-10
**Interesting techniques for improving diffusion models**

**Rating:** 5
**Confidence:** 3

**Review:**

**Summary**

This paper presents rich discussions and various practical techniques to improve the training of probabilistic diffusion models, which include a hybrid objective to learn the variance for improving log-likelihood performance, a different noise schedule tailored for ImageNet 64x64 and importance sampling to reduce gradient noise. Experiments on ImageNet 64x64 and various ablation study provided interesting insights and empirically justified the claims.

**Pros**
The paper is well-written and develops some useful practical techniques for improving a recently proposed deep generative model (a modified version of the diffusion probabilistic model in (Jascha, et al 2015)). Specifically, the paper managed to improve the log-likelihood performance by identifying the issue of the simplified objective and proposed to learn the variance using a hybrid objective. I think this technique along with others are useful practical techniques to improve the training of diffusion models.

**Questions & Concerns**
- No results on CIFAR: Most recent papers in this field considers CIFAR-10 as the standard benchmark to report generative performance, including the original DDPM paper [1] and the score matching paper [2]. Although ImageNet 64x64 is a larger dataset with more complicated structure and diversity, outperforming pervious strong baselines in CIFAR-10 is still challenging and non-trivial. Thus the empirical study will also be more convincing to demonstrate that the proposed method can indeed achieve much better log-likelihood without sacrificing sample quality too much. Otherwise, it's hard to get a sense of how much improvement has been actually achieved by directly looking at the numbers in this paper and the ones in previous papers.

- As the major contribution, the variance parametrization (Eq 16) needs more insightful discussions. For example, why this is the case: "Since Figure 1a shows that the reasonable range for$\Sigma(x_t; t)$ is very small, it is clear that we should not use a neural network to predict$\Sigma(x_t; t)$ directly.". Can we predict the log of the variance with a neural network directly? The proposed one is only an interpolation between $\beta_t$ and $\tilde{\beta}_t$ - is this expressive enough?

- About the noise schedule: is this a generally better noise schedule, or it is only tailored for ImageNet 64x64. In the latter case, I think it is only a trick that overfits a specific dataset. To improve the training of DDPM generally, is there any advice on how to find a good noise schedule?

I will consider raising my score if the above concerns can be addressed.

[1] Denoising Diffusion Probabilistic Models

[2] Generative modeling by estimating gradients of the data distribution

---

> ### Author Response · Authors · 2020-11-20
> **Added CIFAR-10 results, explained variance bounds, clarified schedule choices**
>
> Thank you for taking the time to carefully review our paper! Your feedback is very relevant and insightful, and will help make our paper more clear and insightful to the reader.
>
> * We completely agree with your assessment that CIFAR-10 results will make our case more compelling and aid in comparison to other recent papers. As a result, we have added results for CIFAR-10. We initially avoided CIFAR-10 because it is a low-data regime where overfitting is the main bottleneck, but it has become clear that researchers benefit from testing with such a small and easy-to-use dataset. We hope that adding CIFAR-10 results will demonstrate that our method is more generally beneficial.
>
> * Note that $\beta_t$ and $\tilde{\beta_t}$ are the upper- and lower-bounds on the reverse process variances, as mentioned in Ho et al. These bounds hold because we normalize images into the range [-1, 1], but don’t hold in the general case where the data distribution is not standardized (e.g. if it were something like $N(0, 100)$). As an empirical argument for our parameterization being expressive enough, we never observed the model producing variances outside of $\beta_t$ and $\tilde{\beta_t}$, even though its output $v$ is not clipped to be in the interpolation range [0,1]. We have added this observation to the paper, since we think it will help motivate our choices. We should also mention that Ho et al. predicted the log of the variance (rather than the variance directly), and it did not work well for them (they found it was unstable).
>
> * We believe that our schedule is fairly general (and this is backed up by our results on CIFAR-10, which show improved NLL), but the guidelines that guided our choice were 1) force $\sqrt{\beta_0}$ to be slightly smaller than the size of a discrete pixel bin (i.e. 1/127.5), 2) add noise more slowly so that every step of sampling has an impact on FID. We have updated the paper to clarify these choices. We also believe that the generality of our schedule is more clear with the new CIFAR-10 results.

---

### Author Response · Authors · 2020-11-20
**Added CIFAR-10 and sigma_small results; explain noise schedule & importance of log-likelihood**

First, we would like to thank all reviewers for taking the time to provide us with helpful feedback! The reviews have helped us understand various changes and clarifications we should make to our paper, and will help make this paper stronger.

We’ve updated the paper to include the following:

1. Results on CIFAR-10, where **we improve NLL from 3.70 to 2.94** (Table 3). Our results show that our $L_{hybrid}$ objective, cosine schedule, and variance reduction technique all help NLL on CIFAR-10 with a modest change in FID (Table 2). Additionally, we see that **training with $L_{hybrid}$ allows us to sample with many fewer diffusion steps on this dataset** (Figure 5c), compared to the fixed sigmas from Ho et al., confirming that our ImageNet findings hold for CIFAR-10.
2. Added a discussion on why improving likelihood matters for generative models (Section 3, first paragraph).
3. Incorporated additional explanation for our novel schedule and variance parameterization (Section 3.2).
4. Updated fast sampling results to include comparisons with both $\sigma_{large}$ and $\sigma_{small}$ (Figure 5).

While reading the reviews, we noticed that we under-emphasized one of our key results: that we considerably improve the sampling speed of diffusion models, just by learning variances. Diffusion models are currently slow to sample from because you need to run thousands of forward passes of the model to get a single sample. We show that you can sample with **as few as 50 forward passes** with very little reduction in sample quality, which is very important for the practical use of these models. Furthermore, we found that using our $L_{hybrid}$ objective helps keep sample quality better than using fixed sigmas when sampling with few steps. We have updated the introduction of the paper to put more emphasis on this result.

We’d like to also provide more motivation for why we worked on improving likelihood in our work. For generative models, likelihood is a commonly-used metric for a number of reasons. First, good likelihood is often seen as a proxy for good mode coverage. Second, as we get close to the intrinsic entropy of the distribution, tiny improvements in log-likelihood can lead to drastic changes in sample quality. Earlier work seemed to suggest that diffusion models don’t achieve likelihoods competitive with other models like VAEs, Flows, or auto-regressive models, casting a doubt on whether they truly model the full distribution. However, our work shows that they’re indeed competitive, suggesting that good likelihoods were already hidden in the existing models and that learning sigma helps reveal them. This should give researchers more confidence in these models.

---

### Decision · Program_Chairs · 2021-01-07
**Final Decision**

**Decision:**

Reject

**Comment:**

This paper arose a number of questions and concerns among Reviewers that made it get below-average scores (unfortunately, Reviewers did not provide further feedback on the rebuttal). After discussion between the Program Chairs, calibrating decisions across all submissions and, given the drawbacks mentioned below, it is decided that this paper does not meet the bar for this year's ICLR. Therefore, the final decision is to REJECT the paper. As a brief summary, I highlight below some pros and cons that arose during the review and meta-review processes.

Pros:
- Further developing on a simplification of previous approaches (learning diffusion sigmas).
- Proposal of a new noise schedule.
- Improving the log-likelihood of diffusion-based generative models.
- Improving generation time.

Cons:
- Similar FIDs as non-improved approaches (in some cases).
- Focus on log-likelihood may not be of paramount importance for a generative task.
- Dichotomy between better FID and better NLL could be further discussed.
- More comparison with other approaches and further data sets could be done.
- A bit ad-hoc noise schedule.